# Facile Synthesis of PtPd Network Structure Nanochains Supported on Multi-Walled Carbon Nanotubes for Methanol Oxidation

Dawei Zhang [1,†], Yanrong Ren [2,†], Zhenhua Jin [2], Yonghua Duan [1], Mingli Xu [2] and Jie Yu [1,*]

[1] Faculty of Materials Science and Engineering, Kunming University of Science and Technology, Kunming 650093, China
[2] Faculty of Metallurgical and Energy Engineering, Kunming University of Science and Technology, Kunming 650093, China
* Correspondence: yujie@kust.edu.cn
† These authors contributed equally to this work.

**Abstract:** In this paper, a PtPd NC catalyst with a network structure of nanochains was prepared with KBr as a structure-directing agent, $NaBH_4$ as a reducing agent, and modified multi-walled carbon nanotubes (MWCNTs) as a support. The experimental results show that the structure-directing agent KBr helps to form a particular type of nanochain with a network topology. PtPd NCs with various ratios (Pt:Pd = 2:1, 1:1, 1:2) have respective diameters of 30 nm, 35 nm, and 23 nm. With numerous structural flaws at the junctions, the nanochains' distinctive network structure increases the number of active sites on the PtPd nanocenter surface. Electrochemical characterization results show that the current density of the PtPd NCs is about 658.5 mA $mg^{-1}$, 1.5-times that of the Pt/C catalyst and 3.9-times that of the commercial Pd/C catalyst. Furthermore, it has better electrocatalytic stability for methanol oxidation than Pd/C and Pt/C catalysts.

**Keywords:** network structure; platinum–palladium nanochains; methanol; structural defects; electrocatalysis

## 1. Introduction

Direct methanol fuel cells (DMFCs) have received many researchers' extensive attention due to their high energy density, high conversion efficiency, and environmental friendliness [1,2]. Additionally, due to the abundant supply of methanol, simplicity of storage, and accessibility of current fuel delivery systems, DMFCs have become extremely attractive [3–5]. However, the kinetic process of the methanol oxidation reaction at the DMFC anode is so slow that it produces incomplete oxidation intermediates, such as CO, and these intermediates will make adsorption on the catalyst surface and cover the active site, which will eventually lead to a reduction in catalyst activity. On the other hand, the catalyst will dissolve in the electrolyte during the catalytic process, which will lead to the instability of the whole cell system and affect the service life. Therefore, the development of DMFC catalysts with high catalytic performance and high stability is beneficial for the practical application of DMFCs [6–8].

Compared with the single metal catalysts, Pt/Pd-based catalysts have better stability and catalytic activity [9–13], thus, enhancing the anti-poisoning ability of the catalysts. Studies have shown that special morphologies of PtPd bimetallic nanostructured catalysts present excellent catalytic performance for DMFCs due to the synergistic effect and electronic interaction between Pt and Pd atoms [14,15].

The activity of the catalyst is closely related to its morphology, structure, and particle size [16–21]. Yizhong Lu et al. [22] successfully synthesized PtPd porous nanorods through a bromide-induced galvanic replacement reaction between Pd nanowires and K2PtCl6.

Chunmei Zhang [23] prepared a PtPd alloy with concave nanocubes via a hydrothermal reaction in the presence of GO. Dong and his colleagues [24] used Te nanowires as corrosion templates and reducing agents to prepare AuPtPd nanowires. Guo et al. synthesized Te nanowires via the hydrothermal method, then synthesized Pd nanowires with Te nanochains as a reducing agent and template and, finally, synthesized PtPd nanowires using ascorbic acid as a reducing agent precursor. Achari I. prepared an ultra-thin $Pt_xPd_{(1-x)}$ alloy catalyst by the low potential deposition of a H sacrificial layer surface, restricting redox replacement [25]. However, the reaction time required by the above-synthesized methods is relatively long and the reaction process is very complicated. In addition, these organic structure-directing agents may also hinder the reaction's mass transfer or electron transfer, thus, seriously limiting the practical application of the catalysts. Among various morphologies of PtPd nanocatalysts, PtPd nanochains have a unique network structure [26,27] with a large specific surface area and more active sites that can be used as highly efficient fuel cell catalysts. Multi-walled carbon nanotubes (MWCNTs) have many advantages, such as large specific surface area, good electrical conductivity, and a high degree of crystallization [28–30]. MWCNTs are often used as support for nanoparticles because it is conducive to improving the utilization of precious metals and enhancing the interaction between carriers and noble metal nanoparticles.

In this paper, we use the inorganic reagent potassium bromide as a structure-directing agent to synthesize a PtPd NC catalyst with a network structure at room temperature by a simple one-step chemical reduction method. The morphology, structure, and particle size with electrocatalytic performance were discussed for the synergistic interaction between Pt and Pd.

## 2. Experiments

### 2.1. Chemical Agents

The chemical reagents used are as follows: palladium chloride (II) ($PdCl_2$), potassium chloroplatinate (II) ($K_2PtCl_6$), polyethylene glycol (PEG-400), potassium bromide (KBr), multi-walled carbon nanotubes (MWCNTs), potassium hydroxide (KOH), sodium borohydride ($NaBH_4$), isopropanol, methanol, nafion solution (5 wt%), deionized water (18.25 MΩ cm). All aqueous solutions were prepared with twice-distilled water.

### 2.2. Preparation of PtPd NCs/MWCNTs

Firstly, 24 mL PEG, 1 mL $K_2PtCl_6$ (2.34 mg mL$^{-1}$), and 4 mL $Na_2PdCl_4$ (0.325 mg mL$^{-1}$) were evenly mixed with 1mL KBr (0.1 mol L$^{-1}$). Secondly, 50 mL fresh $NaBH_4$ (0.48 mol L$^{-1}$) was added into the above-mixed solution drop by drop under stirring, then $Pt_1Pd_1$ NCs were obtained after 50 min of reaction under magnetic stirring. The above colloidal solution was added to the previously modified and ultrasonically dispersed MWCNTs (Section 2.1 of the study [4] as the modification method). After stirring for 4 h, $Pt_1Pd_1$ NC/MWCNT catalyst was obtained by filtration and washed with deionized water then dried at 60 °C for 12 h. With the same method as above, $Pt_1Pd_2$ NCs/MWCNTs and $Pt_2Pd_1$ NCs/MWCNTs were synthesized by Pt:Pd metal molar ratio of 1:2 and 2:1, respectively. For comparison, PtPd nanoparticles were also synthesized without structure-directing agent KBr with the same method as the above PtPd NPs.

### 2.3. Physical Characterization

A transmission electron microscope (TEM, Tecnai G2 F20 S-Twin) (FEI, Hillsboro, OR, USA) was used to test the morphology and dispersion of the noble metal nanoparticles operated at 200 kV. A powder X-ray diffractometer (XRD, Rigaku miniFlex600 powder diffractometer) was used to characterize the sample's crystal phase structure and crystal grain size. Diffraction peaks were collected from 10° to 90° (2θ) at a sweep speed of 2° min$^{-1}$. EDS was used to analyze the elements of the catalysts. The chemical components of every element were collected on X-ray Photoelectron spectra (XPS, Thermo K-Alpha$^+$) with a monochromatic aluminum Kα (1486.6 eV) X-ray source operating at 150 W.

*2.4. Electrochemical Measurements*

All electrochemical measurements were performed using a three-electrode system with the CHI660C electrochemical workstation, the glassy carbon electrode as the working electrode, the platinum electrode as the counter electrode, and the saturated calomel electrode (Ag/AgCl) as the reference electrode. The surface of the glassy carbon electrode with a 3 mm diameter was polished to a mirror surface by using $Al_2O_3$ with a diameter of 300 nm and 50 nm in succession, ultrasonically washed in ethanol for 3 min, rinsed with deionized water, and air-dried. Then, 950 μL of isopropanol and 50 μL of Nafion solution were ultrasonically mixed and 2 mg of the prepared catalyst was added and ultrasonicated for another 20 min to form an ink-like suspension. A 10 μL sample was dropped on the surface of a cleaned glassy carbon electrode using a micro-injector and allowed to air dry. The electrocatalytic activity and stability of the catalysts for methanol were tested in a 0.5 M KOH + 2 M $CH_3OH$ solution (the $N_2$ is saturated with the solution for 20 min to remove dissolved oxygen). The electrochemical test temperature was 30 °C, the scan range was −0.8~0.4 V (vs. Ag/AgCl), and the scan rate was 50 mV·s$^{-1}$.

## 3. Results and Discussion

*3.1. Physicochemical Characterization*

Figure 1A,B show the TEM images of $Pd_1Pt_1$ NCs at different magnification ratios (200 nm, 100 nm). The figures display a network structure and rough surface nanochains. This results in a higher specific surface area and more active sites so as to improve catalytic performance. The structural characteristics of $Pt_1Pd_1$NCs are further analyzed by HRTEM. Figure 1C shows that $Pt_1Pd_1$NCs are connected by a stack of nanoparticles with a diameter of about 4.5 nm, easily distinguished by their crystalline interplanar spacing around 0.224 nm (the spacing of Pt and Pd is 0.226 nm and 0.2245 nm). Additionally, the nanoparticles in the nanochains display varying growth directions, which may help the nanochains to develop a network structure.

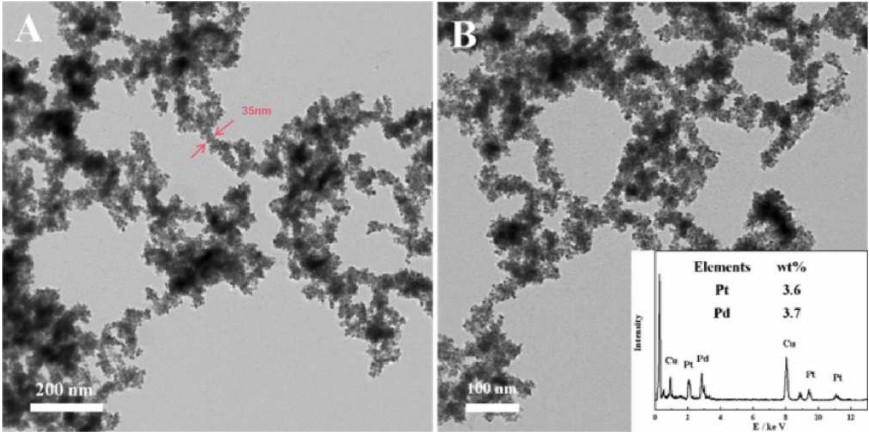

**Figure 1.** *Cont.*

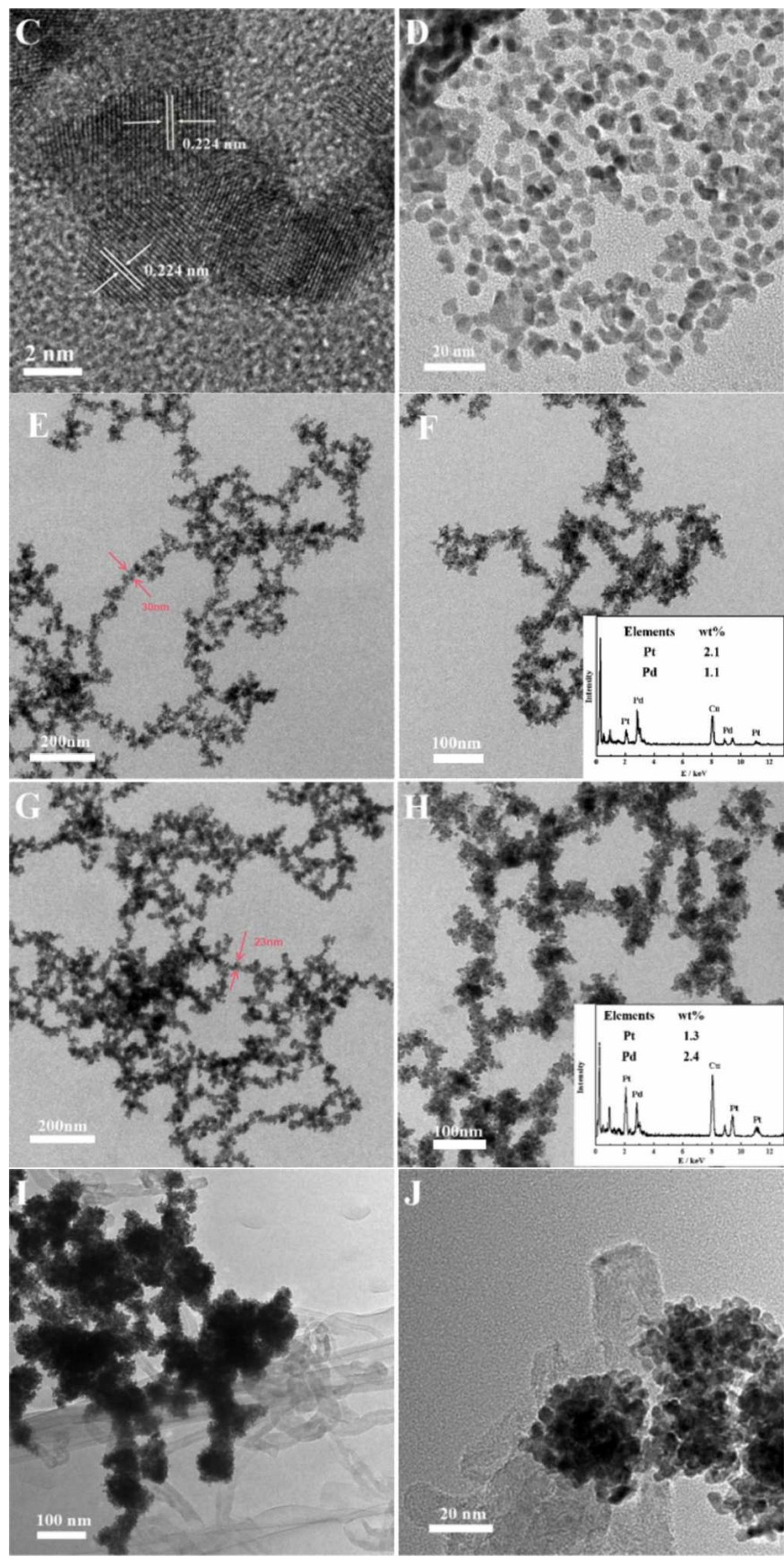

**Figure 1.** The TEM images and EDS of (**A**,**B**) $Pt_1Pd_1$ NCs, (**C**) the HRTEM image of $Pt_1Pd_1$ NCs, (**D**) $Pt_1Pd_1$ NPs (without KBr), (**E**,**F**) $Pt_2Pd_1$ NCs, (**G**,**H**) $Pt_1Pd_2$ NCs, (**I**,**J**) $Pt_1Pd_2$ NCs/MWCNTs.

Figure 1E–H are TEM images of $Pt_2Pd_1$ NCs and $Pt_1Pd_2$NCs at different magnifications (200 nm, 100 nm), respectively. The network structure nanochain's average diameters for $Pt_2Pd_1$ NCs, $Pt_1Pd_1$NCs, and $Pt_1Pd_2$ NCs are about 30 nm, 35 nm, and 23 nm, respectively, and the $Pt_1Pd_2$ NCs have the finest diameter. Because the reduction potential of $Pd^{2+}/Pd$ (0.915 V) is less than that of $Pt^{2+}/Pt$ (1.18 V), Pd may be reduced preferentially. With the increase in Pd content, the nucleation number of the nanoparticles will also increase. When the content of Pd is higher than Pt, the diameter of the nanoparticles becomes small and the nanochains become fine. As a result, tiny nanoparticles are linked together to form nanochains to provide a bigger specific surface area and more active sites, which is advantageous to increase the catalytic performance of the catalyst. For further determining the composition of the nanochains, EDS of $Pt_2Pd_1$NCs, $Pt_1Pd_1$ NCs, and $Pt_1Pd_2$NCs is analyzed. From Figure 1B,F,H (bottom right corner), the nanochains are composed of two elements, Pt and Pd, because the atomic percentage of Pt and Pd in each catalyst is consistent with the molar ratio of metal in the precursor solution, demonstrating that the metal ions in the solution are completely reduced.

Figure 1D shows a TEM image of $Pt_1Pd_1$ NPs without the structure-directing agent KBr for comparison. As Figure 1D shows, a large number of PtPd NPs nanoparticles have formed and there is no network structure. There may be two possible reasons: (1) Since $Br^-$ has a strong chemisorption capacity, it is advantageous as a ligand to form $[PdBr4]^-$ and $[PtBr4]^-$ complexes in PEG-400 with a helical structure [29]. As a protective agent, it is quickly reduced to fine nanoparticles. (2) $Br^-$ can easily combine with the (100) plane of the face-centered cubic crystal structure of the metal to suppress the growth of the (100) crystal plane and $Br^-$ has an etching effect. They jointly lead to the formation of a network structure of PtPd network structure nanochains. In addition, Figure 1I,J show that the $Pt_1Pd_2$ NCs are uniformly supported on the surface of MWCNTs. It is advantageous to increase the specific surface area, form more active sites, and enhance the electrocatalytic performance.

XRD spectra of three as-synthesized PtPd NCs/MWCNTs catalysts are shown in Figure 2. The diffraction angles $2\theta 39.9°$, $46.4°$, $67.8°$, and $81.6°$, which correspond to the (111), (200), (220), and (311) crystal planes of Pt and Pd, respectively, certify that the synthesized nanochains are a face-centered cubic structure [31]. The diffraction peaks of $Pt_2Pd_1$ NCs/MWCNTs, $Pt_1Pd_1$NCs/MWCNTs, and $Pt_1Pd_2$ NCs/MWCNTs located between the standard peaks of Pt (JCPDS No. 04-0802) and Pd (JCPDS No. 46-1043) indicate that Pt and Pd form an alloy structure. As the Pd content increases, the diffraction peak shifts toward the standard peak of Pd, but the trend is not particularly noticeable. This may be because the lattice constants of Pt and Pd are very close, resulting in an insignificant shift tendency.

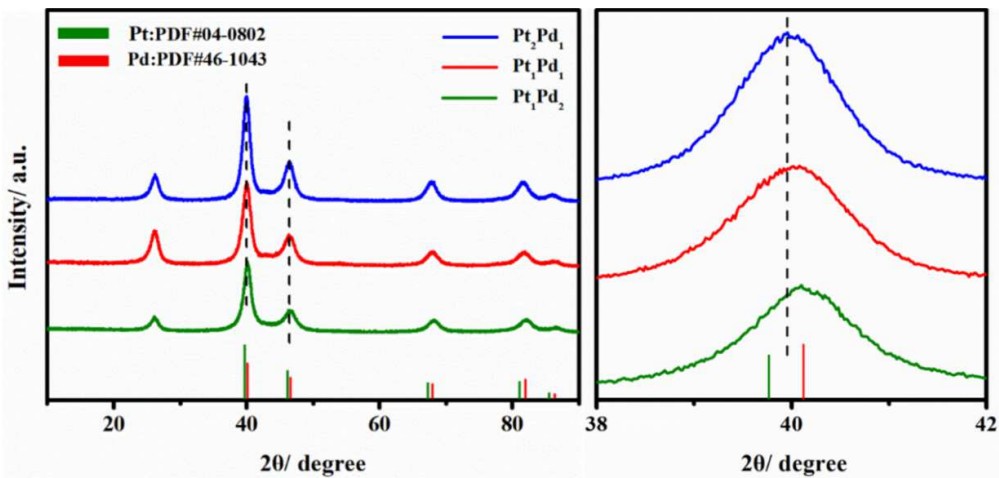

**Figure 2.** XRD patterns of $Pt_2Pd_1$ NCs/MWCNTs, $Pt_1Pd_1$ NCs/MWCNTs and $Pt_1Pd_2$ NCs/MWCNTs.

In order to describe the electronic structure of Pt and Pd in PtPd NCs affected by different molar ratios, the XPS spectra of Pt 4d and Pd 3d of three proportions of PtPd NCs are compared and the electrons corresponding to Pt 4f and Pd 3d are combined. The energy values are listed in Table 1. Figure 3A shows the Pt 4f spectra of $Pt_1Pd_2$ NCs with two major peaks at 71.4 and 74.7 eV for metallic Pt 4f7/2 and 4f5/2, respectively, while other peaks at 72.3 and 75.2 eV show the oxidized form of Pt. Figure 3B shows the Pd 3d spectrum of $Pt_1Pd_2$ NCs, the two main peaks at 335.2 and 341.1 eV correspond to Pd(0), while the other peaks at 336.30 eV and 342.11 eV show the oxidized form of Pd. Based on the peak areas, it is known that Pt(0) and Pd(0) are mainly present in $Pt_1Pd_2$ NCs. Figure 3C is a Pt 4f spectrum of $Pt_1Pd_1$ NCs, $Pt_2Pd_1$ NCs, $Pt_1Pd_2$ NCs, and Pt NCs. For the Pt 4f 7/2 and Pt 4f 5/2 orbits of the Pt NCs catalyst, the electron-binding energies are 71.34 eV and 74.88 eV, respectively. As the Pt ratio of the nanochains decreases, compared with the Pt NCs, the offset of $Pt_2Pd_1$ NCs is maximum and the electron-binding energies of Pt 4f 7/2 and Pt 4f 5/2 are shifted to 0.52 eV and 0.67 eV, respectively. Figure 3D shows the Pd 3d spectra of $Pt_1Pd_1$ NCs, $Pt_2Pd_1$ NCs, $Pt_1Pd_2$ NCs, and Pd NCs. For the Pd 3d 5/2 and Pd 3d 3/2 orbits of the Pd NC catalyst, the electron-binding energy is 335.74 eV and 341.24 eV, respectively. Compared with Pd NCs, $Pt_2Pd_1$ NCs have the largest offset and the electron-binding energies of Pd 3d 5/2 and Pd 3d 3/2 are negatively shifted by 0.67 eV and 0.81 eV, respectively. The offset may be due to the internal electron transfer between Pt and Pd, resulting in a synergistic effect. This feature has a significant impact on the catalytic performance of PtPd NCs. In addition, it also proves that Pt and Pd form an alloy, which is consistent with the XRD results.

**Table 1.** Value of binding energies of Pt and Pd.

| Samples | Pt 4f$_{7/2}$ | Pt 4f$_{5/2}$ | Pd 3d$_{5/2}$ | Pd 4d$_{3/2}$ |
|---|---|---|---|---|
| Pt | 71.34 | 74.88 | - | - |
| Pt:Pd = 2:1 | 70.82 | 74.21 | 335.07 | 340.43 |
| Pt:Pd = 1:1 | 71.18 | 74.44 | 335.22 | 340.60 |
| Pt:Pd = 1:2 | 71.16 | 74.41 | 335.43 | 340.78 |
| Pd | - | - | 335.74 | 341.24 |

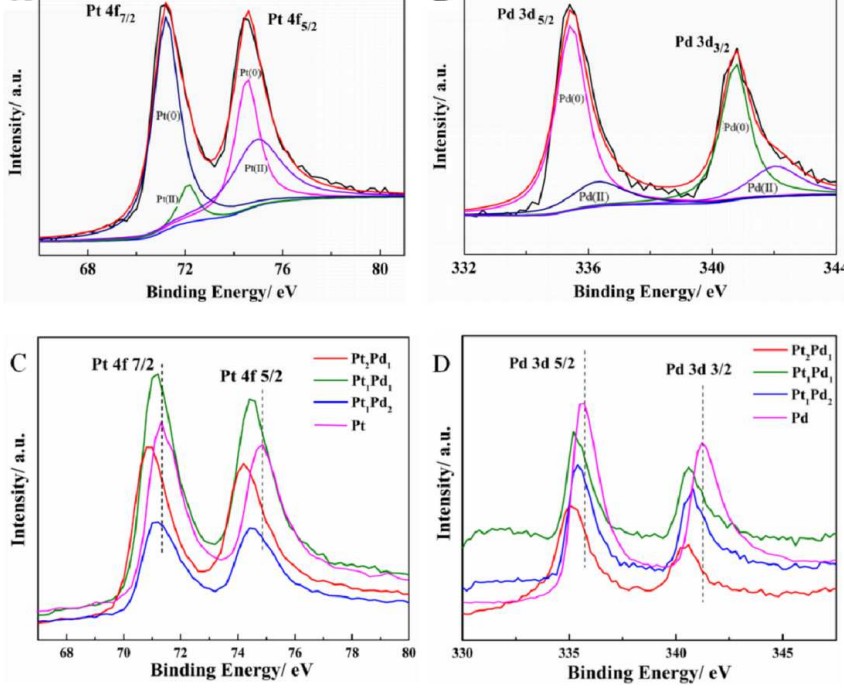

**Figure 3.** XPS spectra of (**A**) Pt 4f, (**B**) Pd 3d of $Pt_1Pd_2$ NCs; and (**C**) Pt 4f; (**D**) Pd 3d of Pt NCs, Pt2Pd1 NCs, Pt1Pd1 NCs, Pt1Pd2 NCs, and Pd NCs.

### 3.2. Electrochemical Characterization

Simultaneously, the electrocatalytic ability of PtPd NCs/MWCNT catalysts for methanol oxidation is studied and compared with the commercial Pd/C catalyst (JM) and the commercial Pt/C catalyst (JM). Figure 4A is a cyclic voltammogram of $Pt_1Pd_1$ NCs/MWCNTs, $Pt_2Pd_1$ NCs/MWCNTs, $Pt_1Pd_2$ NCs/MWCNTs, Pd/C (JM), and Pt/C (JM) catalysts in 0.5 M KOH + 2.0 M $CH_3OH$ solution. It can be seen from Figure 4A that the peak current is located at about 0.9V in the forward scan, which is usually the oxidation current of the C-H bond splitting during methanol oxidation. In reverse scanning peak current is attributed to the oxidation species formed on the catalyst surface in the CV curve, which is about 0.8 V during the reverse scan. In the forward scan, the $Pt_1Pd_2$ NCs/MWCNT catalyst showed the highest current density (658.5 mA $\cdot$ $mg^{-1}$) of all these synthesis catalysts, which was 1.5-times the commercial Pt/C (436 mA $\cdot$ $mg^{-1}$) and 3.9-times the commercial Pd/C (169 mA$\cdot mg^{-1}$). At the same time, $Pt_1Pd_1$-NCs/MWCNT and $Pt_2Pd_1$-CNS/MWCNT catalysts are also higher than commercial Pd/C catalysts and commercial Pt/C catalysts. The high electrocatalytic activity of PtPd NCs/MWPNT catalysts for methanol oxidation may be attributed to the synergistic effect between Pt and Pd atoms. The high catalytic activity of the $Pt_1Pd_2$-NCs/MUCNT catalyst while reducing the Pt content is related to more active sites being exposed by the smaller diameter of $Pt_1Pd_2$ NCs/MWCNT nanochains. In reverse scanning, the peak current density of the $Pt_1Pd_2$-NCs/MWCNT catalyst is still higher than that of $Pt_1Pd_2$ NCs/MWCNT, $Pt_2Pd_1$-NCs/MWPNT, Pt/C, and Pd/C catalysts, which indicates that the $Pt_1Pb_2$-NCs/MWCET catalyst has good reversibility of oxygen adsorption and desorption [32]. In addition, we compared the catalysts reported in other studies for methanol fuel cells and the corresponding comparisons are listed in Table 2. It shows that PtPd NCs/MWCNT catalysts are superior to many other reported electrocatalysts [33–37].

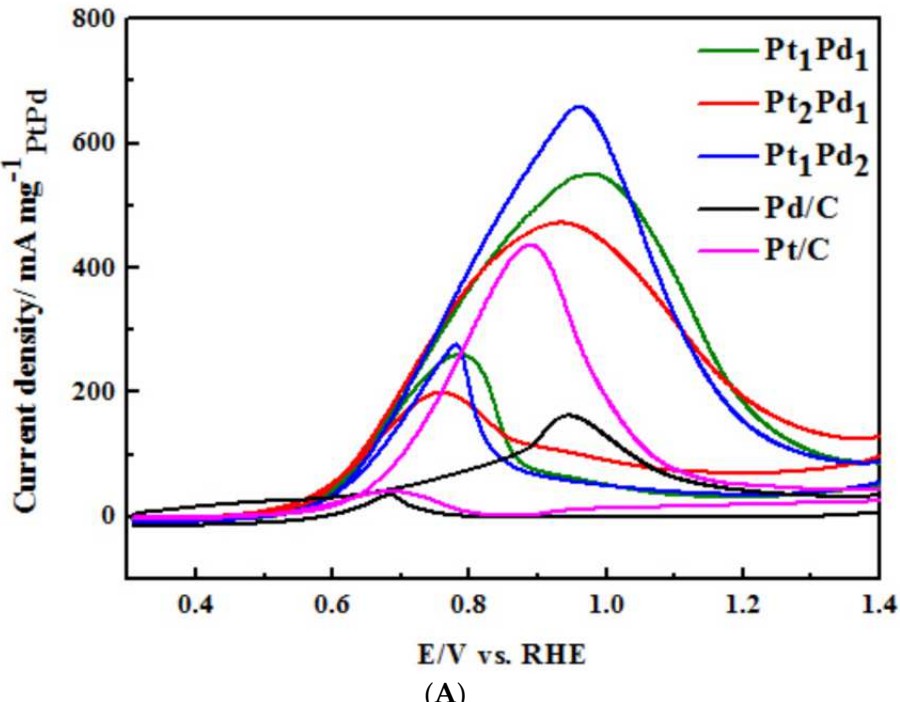

**(A)**

**Figure 4.** *Cont.*

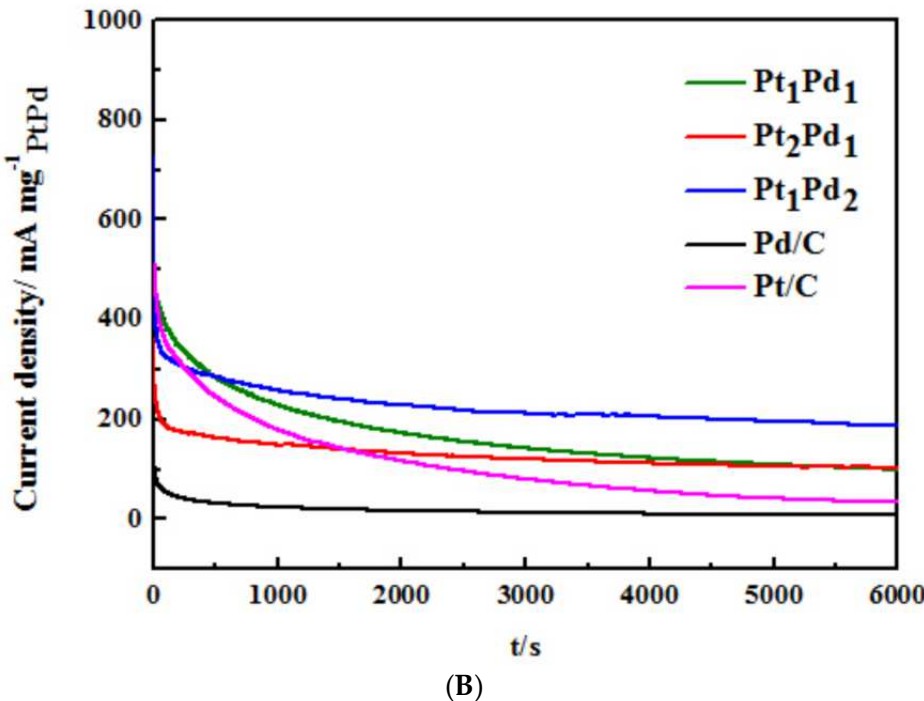

**(B)**

**Figure 4.** (**A**) CV, (**B**) i–t curves of $Pt_2Pd_1$ NCs/MWCNTs, $Pt_1Pd_1$ NCs/MWCNTs, $Pt_1Pd_2$ NCs/MWCNTs and the commercial Pd/C catalysts in a solution containing 0.5 M KOH +2.0 M $CH_3OH$.

**Table 2.** The compared electrochemical performance of PtPd NCs/MWCNTs with the other reported Pd-based catalysts for methanol oxidation.

| Catalysts | Electrolytes | Scan Rate (mV·s$^{-1}$) | Current (mA·mg$^{-1}_{Pd}$) | Ref. |
|---|---|---|---|---|
| Pd/NS-G | 0.5 M NaOH + 1.0 M Methanol | 50 | 399 | [34] |
| Nanoporous Pd/NiO | 0.5 M NaOH + 0.5 M Methanol | 50 | 344 | [35] |
| Pt-MnO$_2$/MWCNTs | 0.5 M NaOH + 1.0 M Methanol | 50 | 431 | [36] |
| YS Pd-Ni-P/C | 0.5 M KOH + 1.0 M Methanol | 50 | 524 | [37] |
| Pt$_{50}$Pd$_{50}$/SCs | 1.0 M KOH + 1.0 M Methanol | 50 | 336 | [38] |
| Pt$_2$Pd$_1$/MWCNTs | 0.5 M KOH + 2.0 M Methanol | 50 | 472 | This work |
| Pt$_1$Pd$_1$/MWCNTs | 0.5 M KOH + 2.0 M Methanol | 50 | 531 | This work |
| Pt$_1$Pd$_2$/MWCNTs | 0.5 M KOH + 2.0 M Methanol | 50 | 685 | This work |

The synergistic effect of the PtPd alloy can not only promote the oxidation of adsorbed intermediate oxygen species at low potential, but also improve the anti-poisoning ability of the catalyst. In the reaction process, *Pt* occupies the main dehydrogenation site and *Pd* accelerates the removal of *CO*, thereby inhibiting the poisoning of *Pt*. The mechanism of the reaction is summarized as follows [38]:

$$CH_3OH + Pt \rightarrow Pt - CO_{ad} + 4H^+ + 4e^- \tag{1}$$

$$Pt - CO_{ad} + Pd \rightarrow Pd - CO_{ad} + Pd \tag{2}$$

$$Pd + H_2O \rightarrow Pd - OH_{ad} + H^+ + e^- \tag{3}$$

$$Pt - CO_{ad} + Pd - CO_{ad} \rightarrow CO_2 + Pt + Pd + H^+ + e^- \tag{4}$$

$$Pd - OH_{ad} + Pd - CO_{ad} \rightarrow 2Pd + CO_2 + H^+ + e^- \tag{5}$$

The stability of the catalyst is a critical issue for the lifetime of DMFCs. Therefore, the amperometric i–t curve is employed to compare the stability of these catalysts, including

the above two commercial catalysts. The current density of the $Pt_1Pd_2$ NCs/MWCNT catalyst shows the slowest decay among the three synthesized catalysts, as shown in Figure 4B, and it may be attributed to the synergistic effect of the alloy formed by different ratios of Pt and Pd, which produces different removal capacity of CO-like intermediates in methanol oxidation. The highest current density throughout the test indicated that $Pt_1Pd_2$ NCs/MWCNTs have the best stability and catalytic activity for methanol electrooxidation. According to the aforementioned electrochemical test results, PtPd has a distinctive network structure, a bigger specific surface area, and more active sites, which improves the electro-catalytic activity of the catalysts. Therefore, modifying the morphology by using the structure-directing agent KBr is crucial for enhancing the catalytic activity.

## 4. Conclusions

In this paper, a simple and effective one-step chemical reduction method is used to synthesize different molar ratios of bimetallic PtPd NC/MWCNT catalysts, with high catalytic activity and very good stability for methanol oxidation. The $Pt_1Pd_2$NCs/MWCNT catalyst has the highest electro-catalytic activity, 1.5-times that of the commercial Pt/C and 3.9-times that of the commercial Pd/C catalyst. It has better electro-catalytic stability than the commercial Pt/C and the commercial Pd/C catalysts. With the structure-directing agent KBr, PtPd NCs form a network structure with rough surface defects, which is beneficial to improve the catalytic activity and stability of the catalyst. In addition, Pt and Pd form the PtPd alloy structure and the intermediate species absorbed on the surface of the catalyst are more easily activated due to the interaction of the electrons between Pt and Pd atoms.

**Author Contributions:** Data curation, D.Z.; Investigation, Y.R.; Methodology, M.X.; Resources, Y.D. and J.Y.; Supervision, Z.J. and J.Y.; Writing—original draft, D.Z.; Writing—review & editing, J.Y. All authors have read and agreed to the published version of the manuscript.

**Funding:** This work was financially supported by the National Natural Science Foundation of China (No. 51764030), the Major Special Project of Yunnan Province (202102AB080007), the Natural Science Foundation of Yunnan Province (202001AS070010), and the Analysis and Testing Foundation of Kunming University of Science and Technology.

**Institutional Review Board Statement:** Not applicable.

**Informed Consent Statement:** Not applicable.

**Data Availability Statement:** Not applicable.

**Conflicts of Interest:** The authors declare no conflict of interest.

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
