# Peer review of "Facile Synthesis of PtPd Network Structure Nanochains Supported on Multi-Walled Carbon Nanotubes for Methanol Oxidation"

_metals, doi:10.3390/met12111911_

Round 1

Reviewer 1 Report

The manuscript metals-1994974 by J. Yu et al is one of the very many papers tackling the established field of methanol oxidation catalysis. The work shows that using a facile method for synthesis of Pd-Pt alloy nanoparticles with different atomic ratios could improve in average about twice the performance methanol oxidation catalysts in terms of activity and durability. The paper is likely publishable in metals but before that the authors are advised to go through a moderate revision process that should take care of the following items:

1. The introduction should be making a clearer point on the current status of the methanol electro-oxidation field by explicitly stating what are the main challenges remaining to be addressed.

2. The discussion of authors' own results and findings should make a substantially stronger and more extensive point on the performance of their catalysts in the tests for activity and durability with discussion of the shape of the CV obtained and their comparison to other leading works in the field. 

3. The authors should pay more attention on work of other groups with nanostructured Pd-Pt alloy materials. In that respect, they may wish to consider for instance certain level of comparison with another paper in a MDPI journal on the performance of Pt-Pd nanoporous metal alloy catalysts (I. Achari and N. Dimitrov, Electrochem, 2020, 1, 4–19).

4. The authors may wish to engage a native English speaker to proof-read one more time the work before submission of the revised version for same additional language improvement.

Author Response

Thank you for comments

Reviewer 2 Report

In this paper, PtPd NCs (nanochains?) catalyst with a network structure nanochains were syn using KBr as a structure-directing agent, NaBH4 as reducing agent, and modified multi-walled carbon nanotubes (MWCNTs) as support. The results show that the structure-directing agent KBr helps the formation of PtPd NCs with a network topology useful as an oxygen-reduction catalyst for potential use in methanol fuel cells.  Electrochemical characterization results show that the current density is about 658.5 mA mg-1, 1.5 times that of the Pt/C catalyst 18 and 3.9 times that of the commercial Pd/C catalyst. It has better electrocatalytic stability for methanol oxidation than Pd/C and Pt/C catalysts. The data have been clearly presented and show its potential application in methanol fuel cells.

Author Response

Thanks for suggestion.
